# Multi-Objective Deep Learning with Adaptive Reference Vectors

**Weiyu Chen**    **James T. Kwok**
Department of Computer Science and Engineering
The Hong Kong University of Science and Technology
Hong Kong
{wchenbx, jamesk}@cse.ust.hk

## Abstract

Many deep learning models involve optimizing multiple objectives. Since objectives are often conflicting, we aim to get diverse and representative trade-off solutions among these objectives. Gradient-based multi-objective optimization (MOO) algorithms using reference vectors have shown promising performance. However, they may still produce undesirable solutions due to mismatch between the pre-specified reference vectors and the problem's underlying Pareto front. In this paper, we propose a novel gradient-based MOO algorithm with adaptive reference vectors. We formulate reference vector adaption as a bilevel optimization problem, and solve it with an efficient solver. Theoretical convergence analysis is also provided. Experiments on an extensive set of learning scenarios demonstrate the superiority of the proposed algorithm over the state-of-the-art.

## 1 Introduction

Deep learning models are often evaluated under multiple, potentially conflicting, criteria. For example, in multi-task learning [6], a single model is required to perform well on multiple tasks. In some scenarios, besides accuracy, model fairness is also important so as to ensure that the model is not biased against gender and race. These problems can all be formulated as multi-objective optimization (MOO) [39] problems, and have attracted attention from various fields such as energy resource optimization [9] and signal processing [4].

Since the multiple objectives usually cannot be optimized simultaneously by a single solution, the goal of MOO is to find a set of solutions with different trade-offs to approximate the true Pareto front (PF). The past decades have witnessed the birth of a large number of gradient-free MOO algorithms, such as evolutionary multi-objective optimization algorithms [11, 52] and Bayesian multi-objective algorithms [26, 3]. These algorithms perform well on small-scale problems but fail to provide useful solutions when facing the huge number of parameters in deep learning models.

Recently, gradient-based MOO algorithms [14] have demonstrated promising performance in deep learning. Following the pioneering work in [45], Lin et al. [31] and Mahapatra et al. [37] propose to use a set of reference vectors (RV), and generate multiple solutions on the PF each of which is closest to an RV. Later, strategies are further proposed to improve the efficiency by training a hypernetwork [40, 30] or reference-vector-conditioned network [44]. However, note that a set of uniformly distributed RVs may not lead to a set of uniformly distributed solutions. Hence, an important limitation of these algorithms is that solutions generated using fixed RVs may not cover some parts of the PF, thus failing to provide enough information about the PF. As an illustration, in Figure 1a, solutions obtained by the fixed RVs are close to the ends of the PF. If the RVs are properly positioned, it is possible that the obtained solution set can uniformly cover the PF (Figure 1b).

36th Conference on Neural Information Processing Systems (NeurIPS 2022).

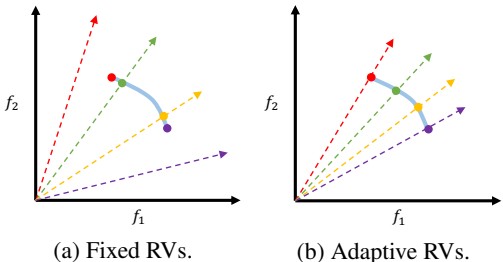

(a) Fixed RVs.  (b) Adaptive RVs.

Figure 1: Illustration of the difference between fixed and adaptive reference vectors. The dashed lines are reference vectors, the blue curve is the PF, and the points are solutions on the PF closest to each reference vector.

Since it is impossible to know the true PF of the problem before optimization, how to generate a proper set of RVs is a big challenge. In this paper, we propose to learn the set of RVs simultaneously with the model parameters. This is formulated as a bilevel optimization problem [13], in which the lower-level optimization problem obtains the Pareto-optimal solutions with a given set of RVs, while the upper-level problem optimizes the RVs based on a quality measure. Moreover, we solve this optimization problem with an inexpensive solver, while still showing theoretically nice convergence properties. Experiments on an extensive set of learning scenarios demonstrate the superiority of the proposed algorithm.

**Notations.** In this paper, $[m]$ denotes the set $\{1, \ldots, m\}$, and $\mathbb{R}_+^m \equiv \{u \in \mathbb{R}^m | u_i \geq 0, \forall i \in [m]\}$. For two vectors $u, v \in \mathbb{R}^m$, $u \geq v$ means that $u_i \geq v_i$ for all $i \in [m]$. $\| \cdot \|$ denotes the Euclidean norm for vectors and the Frobenius norm for matrix.

## 2 Related Work: Multi-Objective Optimization (MOO)

In multi-objective optimization (MOO) [39], one aims to minimize $m \geq 2$ objectives $\{f_1(\phi), f_2(\phi), \ldots f_m(\phi)\}$, or, equivalently, the vector-valued function:

$$\min_{\phi} f(\phi) = [f_1(\phi), \ldots, f_m(\phi)] \in \mathbb{R}^m. \tag{1}$$

A solution $\phi_1$ is *dominated* by another solution $\phi_2$ if and only if $f_i(\phi_1) \geq f_i(\phi_2)$ for $i \in [m]$, and $\exists i \in [m], f_i(\phi_1) > f_i(\phi_2)$. A solution $\phi^*$ is *Pareto-optimal* if and only if it is not dominated by any other $\phi'$. A *Pareto front* (PF) is the set of multi-objective values of all Pareto-optimal solutions. A PF is *regular* [48] if it is simplex-like (i.e., all vectors with positive directions intersect it when they start at the origin), and irregular otherwise. It is shown that regular PFs are not very realistic [24].

Since the number of Pareto-optimal solutions is usually large or even infinite, a set of $n$ Pareto-optimal solutions $\Phi = \{\phi_1, \ldots, \phi_n\}$ is often used to approximate the PF. Let the corresponding multi-objective values be $\mathcal{F} = \{f(\phi_1), \ldots, f(\phi_n)\}$. The quality of $\Phi$ can be evaluated from two perspectives: convergence and diversity [27]. Convergence refers to the distance between $\mathcal{F}$ and the true PF, while diversity measures whether the solutions are well-distributed in the space of objectives. A popular measure, which evaluates both convergence and diversity, is the hypervolume (HV) [53]. Given a reference point $z \in \mathbb{R}^m$, the HV of $\mathcal{F}$ is:

$$HV(\mathcal{F}; z) = \lambda(\cup_{f_i \in \mathcal{F}}[z, f_i]), \tag{2}$$

where $[z, f_i] \equiv \{q \in \mathbb{R}^m | f_i \leq q \leq z\}$, and $\lambda(\cdot)$ is the Lebesgue measure of a set.

A reference vector (RV) [52], sometimes called the weight vector or preference vector, can be used to guide the optimization algorithm by indicating the preferred point on the PF. Usually, an algorithm is expected to obtain the Pareto-optimal solution closest to the given RV in the objective space. Given a RV $r \in \mathbb{R}_+^m$, the MOO problem in (1) can be converted to a single-objective problem by using a scalarization function $s(\phi; r)$. The most straightforward construction is linear scalarization:

$$s(\phi; r) = \sum_{j=1}^{m} r(j) f_j(\phi), \tag{3}$$

which weights the $m$ objectives with the elements $r(j)$'s of $r$. It is known that the minimizer $\phi$ of $s(\phi; r)$ is also Pareto-optimal for the original problem in (1) (Ch 4.7, [5]). However, the vector of objectives $f(\phi)$ obtained may be far away from the given RV. To encourage them to be closer to this vector, a penalty term can be added to the linear scalarization function, leading to [44]:

$$s(\phi; r) = \sum_{j=1}^{m} r(j) f_j(\phi) + \gamma \frac{r^\top f(\phi)}{\|r\| \cdot \|f(\phi)\|}, \tag{4}$$

where $\gamma > 0$ is a constant. Given a (discrete) set of solutions, Ma et al. [36] generate exploration directions to spawn new solutions on the PF, leading to a continuous PF.

By using $n$ RVs from a subset $\mathcal{R} \subseteq \mathbb{R}_+^{m \times n}$, a set of solutions can be obtained to approximate the entire PF. Usually, $\mathcal{R}$ is simply set to $\mathbb{R}_+^{m \times n}$. However, sometimes the decision-makers may only be interested in a specific region of the PF [2]. For example, we may limit the angle between any RV and each coordinate axis. $\mathcal{R}$ can then be changed to:

$$
\left\{ \{r_1, \ldots, r_n\} \;\middle|\; \cos \varphi_2 \leq \frac{r_j^\top u_i}{\|r_j\| \cdot \|u_i\|} \leq \cos \varphi_1, \; \forall i \in [m], j \in [n] \right\}, \tag{5}
$$

where $u_i$ is the $i$th coordinate axis and $\varphi_1, \varphi_2$ are the maximum and minimum allowable angles, respectively. However, note that the set of pre-specified RVs may not fit the problem's PF (e.g., some of them may not intersect the underlying PF), leading to an undesirable solution distribution.

**Gradient-Free MOO.** Evolutionary MOO algorithms (e.g., NSGA-II[11], MOEA/D[52]) and Bayesian MOO algorithms (e.g., BMOA[26], USeMO[3]) are widely used for small-scale black-box problems. These algorithms assume that gradient information is not available. Hence, they often fail to converge on deep learning problems where the solution space can be very large.

**Gradient-Based MOO.** Gradient-based MOO algorithms are more efficient when problems have differentiable objectives and a large number of parameters. Sener and Koltun [45] propose to apply Multiple-Gradient Descent Algorithm (MGDA) [14] to multi-task learning. Liu and Vicente [33] provide theoretical analysis of stochastic MGDA. MGDA can be further extended to incorporate RVs (e.g., EPO [37, 38] and Pareto-MTL [31]). Some algorithms (e.g., MOO-SVGD [34], HIGA [49]) can output a solution set without using RVs. However, they optimize several neural networks simultaneously, and so are computationally expensive and need large GPU memory (especially when the neural network is large).

## 3 Proposed Algorithm

As mentioned in Section 1, the fixed uniformly-distributed RVs used in common practice may result in undesirable solution distributions. Instead of using a fixed set of $n$ RVs ($R = [r_1, \ldots, r_n] \in \mathcal{R}$), we propose to adapt them so that the resultant solution set is well-distributed. In Section 3.1, we first introduce a reference vector-conditioned neural network so that reference vectors can be easily handled without using a lot more parameters. Section 3.2 then formulates reference vector adaptation as a bilevel optimization problem, and an efficient solver is presented in Section 3.3. Its convergence properties are then studied in Section 3.4

### 3.1 Reference Vector-Conditioned Neural Network

In deep learning models, $\phi$ corresponds to the network parameters. Optimizing $\Phi = \{\phi_1, \ldots, \phi_n\}$ means optimizing $n$ neural networks (as in EPO [37] and MOO-SVGD [34]), which is highly inefficient. To alleviate this problem in deep MOO algorithms, Navon et al. [40] and Lin et al. [30] propose to train a single hypernetwork [22] that can output neural network parameters based on the RV. In particular, the Pareto Hypernetwork (PHN) in [40] proposes two ways to optimize the hypernetwork: (i) PHN-LS, which uses linear scalarization, and (ii) PHN-EPO, which uses EPO. However, the hypernetwork still incurs significant computational overhead. For example, the hypernetwork in [40] is around 100 times larger than the base neural network. In this paper, we use the more efficient conditioned network [15, 44]. Specifically, COSMOS [44] concatenates the RV and data sample, and treats this as a joint input to the network. On the other hand, YOTO [15] is originally developed for use with a family of parameterized loss functions, in which the loss parameter is incorporated with the sample into the network via FiLM layers [42]. It is theoretically shown that YOTO is as powerful as using $n$ neural networks [15]. In this work, we adapt the YOTO architecture by replacing the loss parameter with the RV. In this way, the proposed model only has a small parameter overhead compared to the single deep network.

### 3.2 Reference Vector Adaption via Bilevel Optimization

Consider an RV-conditioned neural network $f(\phi; r)$ with parameter $\phi$ and RV $r$ as input. With a set of RVs $R = [r_1, \ldots, r_n]$, the multi-objective values of a solution set can be written as

$[f(\phi; r_1), \dots, f(\phi; r_n)]$. We use a function $\hat{Q}(\cdot)$ to measure its quality. Two choices are considered in this paper. The first one encourages the $f(\phi; r_i)$'s to be far away from each other (and thus more uniformly distributed in the space of objectives):

$$\hat{Q}(R, \phi) = -\sum_{i,j=1}^{n} \exp\left(-\frac{1}{h^2} \|f(\phi; r_i) - f(\phi; r_j)\|^2\right), \tag{6}$$

where $h$ is a constant. The second one is:

$$\hat{Q}(R, \phi) = HV(\{f(\phi; r_1), \dots, f(\phi; r_n)\}; z), \tag{7}$$

which encourages the maximization of HV in (2). Note that the HV-optimal solution is usually not uniformly distributed [21, 47].

To obtain the set of RVs $R$ that generates $\phi$, we formulate it as a bilevel optimization problem [13]. Recently, bilevel optimization has gained great popularity in many machine learning problems such as meta-learning [18], neural architecture search [32], and hyperparameter optimization [19]. We consider the following bilevel optimization problem:

$$\min_{R \in \mathcal{R}} \quad Q(R, \phi^*(R)) \tag{8}$$
$$\text{s.t.} \quad \phi^*(R) = \arg\min_{\phi} S(R, \phi), \tag{9}$$

where $Q(R, \phi^*(R)) \equiv -\hat{Q}(R, \phi^*(R))$, $S(R, \phi) \equiv \sum_{i=1}^{n} s(\phi; r_i)$, and $s(\phi; r)$ is the scalarization function in (3) or (4).[1] The lower-level optimization problem (9) obtains the Pareto-optimal solutions with the given set of RVs $R$, while the upper-level optimization problem (8) optimizes $R$ to maximize the corresponding solution quality.

The idea of RV adaption is also used in some evolutionary algorithms [43, 29]. However, they update the RVs using information from the current population and archive, and cannot be directly used in gradient-based MOO algorithms. Moreover, they cannot scale to problems with a large number of parameters, as is typically the case in deep learning.

### 3.3 Solving the Bilevel Optimization Problem

There are various bilevel optimization solvers for (8). Many of them involve propagation through the inner loop [18, 20], which has a large computational overhead compared to [44]. The proposed algorithm, which will be called Gradient-based Multi-Objective Optimization with Adaptive Reference vectors (GMOOAR), is shown in Algorithm 1. It performs only one stochastic gradient descent step in both the inner and outer loops, as in FO-MAML [18] and DARTS [32]. In each iteration $k$, mini-batches $\xi_k$ and $\pi_k$ are randomly sampled from the data and then used to estimate the stochastic gradients $\nabla_\phi S(R_k, \phi_k; \xi_k)$ and $\nabla_R Q(R_k, \phi_{k+1}; \pi_k)$, respectively. $\text{proj}_{\mathcal{R}}(\cdot)$ is the Euclidean projection operator onto $\mathcal{R}$ that ensures that RVs are inside $\mathcal{R}$. In the sequel, the algorithm using the uniformity-related quality function (6) will be denoted GMOOAR-U, while the one using HV in (7) will be denoted GMOOAR-HV. The gradient of (6) can be computed directly from the computation graph, while the gradient of (7) can be computed using an efficient dimension-sweeping algorithm [17].

The proposed algorithm has $O(w + m)$ memory and $O(w + m)$ time complexity per iteration, where $w$ is the dimension of $\phi$. Since $m \ll w$ in most cases, it has comparable time and space complexity with COSMOS, which is $O(w)$. In Section 3.4, we will show that this simple solver can still provide theoretical guarantees on its obtained solution.

---

**Algorithm 1** Gradient-based Multi-Objective Optimization with Adaptive Reference vectors (GMOOAR).

---

**Input:** learnable RVs $R$, learning rates $\{\alpha_k, \beta_k\}$, initial parameter $\phi$, number of iterations $K$.
1: **for** $k = 1$ to $K$ **do**
2:     sample a mini-batch $\xi_k$ of samples;
3:     $\phi_{k+1} \leftarrow \phi_k - \beta_k \nabla_\phi S(R_k, \phi_k; \xi_k)$; /* optimize network parameters */
4:     sample a mini-batch $\pi_k$ of samples;
5:     $R_{k+1} \leftarrow \text{proj}_{\mathcal{R}}(R_k - \alpha_k \nabla_R Q(R_k, \phi_{k+1}; \pi_k))$; /* optimize reference vectors */
6: **end for**

---

[1]As we use the RV-conditioned network here, $f_j(\phi)$ (resp. $f(\phi)$) in (3), (4) becomes $f_j(\phi; r)$ (resp. $f(\phi; r)$).

## 3.4 Convergence

In this section, we provide convergence analysis for Algorithm 1. As in [23], we make the following assumptions on $S(R, \phi)$ and $Q(R, \phi)$.

**Assumption 1.** *(i) $S(R, \phi)$ is twice-differentiable in $(R, \phi)$. (ii) $\nabla_\phi S(R, \phi)$, $\nabla^2_{R\phi} S(R, \phi)$, $\nabla^2_{\phi\phi} S(R, \phi)$, $\nabla_R Q(R, \phi)$, and $\nabla_\phi Q(R, \phi)$ are Lipschitz continuous w.r.t. $\phi$ with constants $L_s$, $L_{s,1}$, $L_{s,2}$, $L_{q,1}$, and $L_{q,2}$, respectively. (iii) $\nabla^2_{R\phi} S(R, \phi)$, $\nabla^2_{\phi\phi} S(R, \phi)$, and $\nabla_\phi Q(R, \phi)$ are Lipschitz continuous w.r.t. $R$ with constants $L_{s,3}$, and $L_{s,4}$, and $L_{q,3}$, respectively. (iv) $S(R, \phi)$ is $\mu_s$-strongly convex in $\phi$. (v) $\|\nabla^2_{R\phi} S(R, \phi)\| \leq C_s$ and $\|\nabla_\phi Q(R, \phi)\| \leq C_q$.*

Given $\phi^*(R)$, the gradient of the upper-level objective $u(R) \equiv Q(R, \phi^*(R))$ can be obtained as

$$\nabla u(R) = \nabla_R Q(R, \phi^*(R)) - \nabla^2_{R\phi} S(R, \phi^*(R))[\nabla^2_{\phi\phi} S(R, \phi^*(R))]^{-1} \nabla_R Q(R, \phi^*(R)), \quad (10)$$

where $\nabla_R Q(R, \phi^*(R))$ is the direct gradient with respect to $R$. In [23], $\nabla u(R)$ is evaluated using a surrogate constructed by replacing $\phi^*(R)$ in (10) with $\phi_{k+1}$:

$$\overline{\nabla}_R Q(R_k, \phi_{k+1}) \equiv \nabla_R Q(R_k, \phi_{k+1}) - \nabla^2_{R\phi} S(R_k, \phi_{k+1})[\nabla^2_{\phi\phi} S(R_k, \phi_{k+1})]^{-1} \nabla_R Q(R_k, \phi_{k+1}).$$
$$(11)$$

However, this involves computing the Hessian and is expensive. On the other hand, the proposed algorithm uses $h_q^k \equiv \nabla_R Q(R_k, \phi_{k+1}; \pi_k)$, which is the stochastic estimate of $\nabla_R Q(R_k, \phi_{k+1})$. Note that $\nabla_R Q(R_k, \phi_{k+1})$ is the first-order approximation of the gradient in (11). Such an approximation is also used in [18, 32]. It greatly reduces the time and space complexities, but leads to a bias that can be bounded by a constant:

$$\|\overline{\nabla}_R Q(R_k, \phi_{k+1}) - \mathbb{E}_{\pi_k}[h_q^k]\| = \|\nabla^2_{R\phi} S(R_k, \phi_{k+1})[\nabla^2_{\phi\phi} S(R_k, \phi_{k+1})]^{-1} \nabla_R Q(R_k, \phi_{k+1})\|$$
$$\leq C_s C_q / \mu_s.$$

Next, we also make the following assumption similar to [23]. Let $h_s^k \equiv \nabla_\phi S(R_k, \phi_k; \xi_k)$.

**Assumption 2.** *For any $k \geq 0$, there exist constants $\sigma_s, \sigma_q$, and $b_q$ such that:*

$$\mathbb{E}_{\xi_k}[h_s^k] = \nabla_\phi S(R_k, \phi_k), \quad \mathbb{E}_{\pi_k}[h_q^k] = \overline{\nabla}_R Q(R_k, \phi_{k+1}) + B_k, \quad \|B_k\| \leq b_q,$$
$$\mathbb{E}_{\xi_k}[\|h_s^k - \nabla_\phi S(R_k, \phi_k)\|^2] \leq \sigma_s^2 (1 + \|\nabla_\phi S(R_k, \phi_k)\|^2),$$
$$\mathbb{E}_{\pi_k}[\|h_q^k - B_k - \overline{\nabla} Q(R_k, \phi_{k+1})\|^2] \leq \sigma_q^2.$$

Let $\Delta_\phi^k \equiv \mathbb{E}_{\xi_k}\|\phi_k - \phi^*(R_{k-1})\|$, the expected gap between $\phi_k$ and the optimal network parameter given reference vectors $R_{k-1}$. Similarly, denote the expected gap between $R_k$ and the optimal reference vectors $R^*$ in (8) by $\Delta_R^k \equiv \mathbb{E}_{\pi_k}\|R_k - R^*\|$.

**Theorem 1.** *Assume further that $u(R)$ is $\mu_q$-strongly convex, and the step sizes $(\alpha_k, \beta_k)$ satisfy*

$$\alpha_k \leq \min\left\{c_0 \beta_k^{3/2}, \frac{1}{\mu_q}\right\}, \quad \frac{\beta_{k-1}}{\beta_k} \leq 1 + \beta_k \mu_s / 8, \quad \frac{\alpha_{k-1}}{\alpha_k} \leq 1 + 3\alpha_k \mu_q / 4, \quad (12a)$$

$$\beta_k \leq \min\left\{c_1 \alpha_k^{2/3}, \frac{1}{\mu_s}, \frac{\mu_s}{L_s^2(1 + \sigma_s^2)}, \frac{\mu_s^2}{48 c_0^2 L^2 L_q^2}\right\}, \quad 8\mu_q \alpha_k \leq \mu_s \beta_k, \forall k \geq 0, \quad (12b)$$

*where $L, L_q$ are constants and $c_0, c_1 > 0$ are free parameters. For any $k \geq 1$, the iterates generated by Algorithm 1 satisfy*

$$\Delta_R^k \lesssim \left[\prod_{i=0}^{k-1}(1 - \alpha_i \mu_q)\right]\left[\Delta_R^0 + \frac{L^2}{\mu_q^2}\Delta_\phi^0\right] + \frac{c_1 L^2}{\mu_q^2}\left[\frac{\sigma_s^2}{\mu_s} + \frac{c_0^2 L_q^2}{\mu_s^2}\tilde{\sigma}_q^2\right]\alpha_{k-1}^{2/3} + \frac{b_q^2}{\mu_q^2},$$

$$\Delta_\phi^k \lesssim \left[\prod_{i=0}^{k-1}(1 - \beta_i \mu_s / 4)\right]\Delta_\phi^0 + \left[\frac{\sigma_s^2}{\mu_s} + \frac{c_0^2 L_q^2}{\mu_s^2}\tilde{\sigma}_q^2\right]\beta_{k-1}, \quad (13)$$

*where $\lesssim$ denotes that numerical constants are omitted.*

With diminishing step sizes $\alpha_k = c_\alpha/(k + k_\alpha)$, $\beta_k = c_\beta/(k + k_\beta)^{2/3}$, where

$$k_\alpha = \max\left\{35\left(\frac{L_s}{\mu_s}\right)^3(1 + \sigma_s^2)^{\frac{3}{2}}, \frac{(512)^{\frac{3}{2}}L^2L_q^2}{\mu_q^2}\right\}, \ c_\alpha = \frac{8}{3\mu_q}, \ k_\beta = \frac{k_\alpha}{4}, \ c_\beta = \frac{32}{3\mu_s},$$

the following Corollary shows that the assumptions (12a) and (12b) in Theorem 1 are satisfied, and thus $\Delta_\phi^k$ converge to 0, i.e., $\phi_k$ converges to the optimal solution $\phi^*(R_{k-1})$.

**Corollary 1.** *$\phi_k$ converges to the optimal $\phi^*(R_{k-1})$ of (9).*

## 4 Experiments

In this section, extensive experiments are performed, including synthetic problems (Section 4.1), multi-task learning (Section 4.2), accuracy-fairness trade-off (Section 4.3), and usage on larger networks (Section 4.4). Finally, ablation study is presented in Section 4.5. All experiments are conducted on an RTX-2080Ti with 11GB memory.

### 4.1 Synthetic Problems

Experiments are performed on four commonly used multi-objective benchmark problems [11, 52, 34] with different numbers of objectives: (i) 2-objective DTLZ2 [12], (ii) 3-objective DTLZ2 [12], (iii) 2-objective scaled-DTLZ2 [12], and (iv) 3-objective MaF1 [8]. Their detailed definitions are in Appendix B.1. The PFs of problems (i) and (ii) are regular, while those of (iii) and (iv) are irregular. The number of inputs is set to 30. We aim to get 15 non-dominated solutions for each 2-objective problem, and 36 non-dominated solutions for each 3-objective problem.

The proposed algorithm[2] (GMOOAR-U using quality function (6) and GMOOAR-HV using quality function (7)) is compared with the state-of-the-art COSMOS [44],which uses fixed reference vectors. For the 2-objective problems, reference vectors for COSMOS are generated by following their strategy in [44]. For the 3-objective problems, we generate reference vectors for COSMOS by the method in [10], which can obtain more uniform reference vectors. For GMOOAR, the reference vectors are initialized randomly. As in [40], a neural network (with 2 hidden layers, each with 20 units) is used. More experimental details can be found in Appendix B.2.

Figure 2 shows the solutions and HV values obtained. For the 2-objective and 3-objective DTLZ2 (Figures 2a and 2b), the solutions obtained by COSMOS are relatively uniform. However, on the scaled-DTLZ2 (Figure 2c) and MaF1 (Figure 2d) with irregular PFs, many of its solutions are near the boundary and close to each other. In contrast, solutions obtained by GMOOAR-U are distributed more uniformly. Solutions obtained by GMOOAR-HV are not uniform, but have higher HV than COSMOS and GMOOAR-U.

Figure 3 shows the solution sets obtained by GMOOAR-U on the 3-objective DTLZ2 when the region of interest is a subspace $\mathcal{R}$ constrained as in (5). Note that imposing this constraint on COSMOS is difficult as (i) it is hard to generate a set of uniformly distributed RVs; and (ii) even with a set of uniformly distributed RVs, they may not lead to a uniformly distributed set of solutions.

### 4.2 Multi-Task Learning

In this experiment, we use three benchmark datasets from [31]: Multi-MNIST, Multi-Fashion, and Multi-Fashion+MNIST. In Multi-MNIST, each image is constructed by putting two different MNIST images together, one at the bottom-right (BR) and the other at the top-left (TL). Similarly, Multi-Fashion images are constructed by combining images from FashionMNIST [50], while Multi-Fashion+MNIST images are constructed by combining one MNIST image with one FashionMNIST image. More details can be found in [31]. The goal is to classify both the BR and TL images correctly, by minimizing the two cross-entropy losses using a single neural network. As in [31, 37, 44], we use the LeNet [28] with multi-head as base network. Details of the reference-vector-conditioned network are in Appendix B.3.

We compare the proposed algorithms (GMOOAR-U and GMOOAR-HV) with (i) EPO [37], (ii) Pareto hypernetworks (PHN-LS and PHN-EPO) [40], (iii) MOO-SVGD [34], and (iv) COSMOS

---

[2]We use linear scalarization with penalty term in (4).

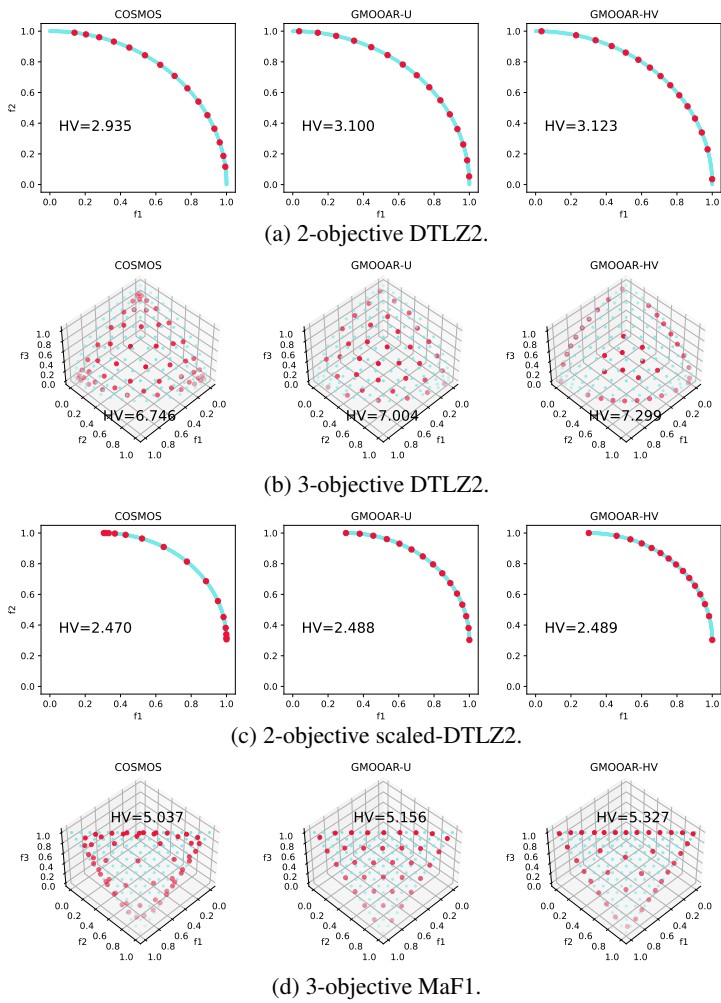

(a) 2-objective DTLZ2.

(b) 3-objective DTLZ2.

(c) 2-objective scaled-DTLZ2.

(d) 3-objective MaF1.

Figure 2: Solution sets (red) and HV values obtained on the synthetic datasets. The Pareto-optimal solutions is in blue.

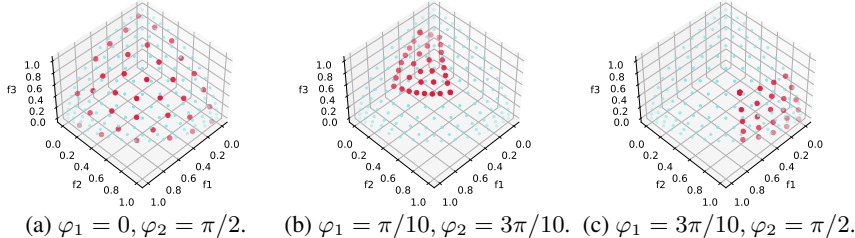

(a) $\varphi_1 = 0, \varphi_2 = \pi/2$.  (b) $\varphi_1 = \pi/10, \varphi_2 = 3\pi/10$.  (c) $\varphi_1 = 3\pi/10, \varphi_2 = \pi/2$.

Figure 3: Solution sets obtained by GMOOAR-U with different $(\varphi_1, \varphi_2)$ settings.

[44]. For EPO, PHN-LS, PHN-EPO and COSMOS, we generate reference vectors following the strategy in [44]. For GMOOAR, the reference vectors are initialized randomly. The experiment is repeated 10 times with different random seeds.

Following common practice [44], we obtain a set of $n$ solutions in each iteration (with $n = 15$ in all experiments). They are evaluated on the validation set every 5 epochs. We only keep the solutions of iteration $k_{best}$ as the final solution set, where $k_{best}$ is the iteration that yields the solution set with the largest validation HV. Note that the original implementation of MOO-SVGD (obtained from the

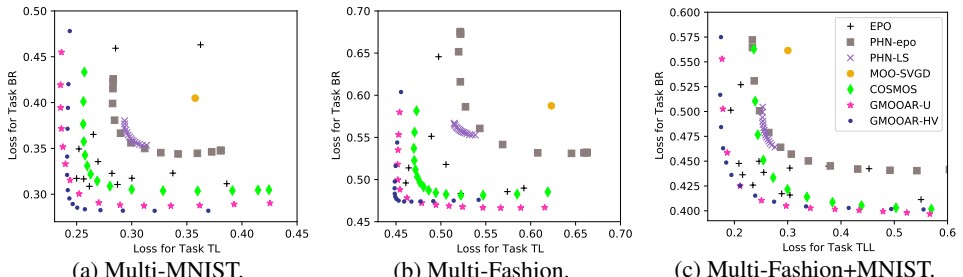

Figure 4: BR and TL test losses obtained on the multi-task learning datasets.

| | HV | | | # parameters | runtime (s) |
| | Multi-MNIST | Multi-Fashion | Fashion-MNIST | | |
|---|---|---|---|---|---|
| EPO [37] | 2.95±0.02 | 2.31±0.01 | 2.86±0.02 | 478,650 | 8,370 |
| PHN-EPO [40] | 2.82±0.04 | 2.16±0.05 | 2.74±0.05 | 3,243,410 | 1,515 |
| PHN-LS [40] | 2.79±0.04 | 2.14±0.04 | 2.67±0.06 | 3,243,410 | 644 |
| MOO-SVGD [34] | 2.67±0.02 | 2.02±0.02 | 2.54±0.04 | 478,650 | 8,661 |
| COSMOS [44] | 2.95±0.02 | 2.31±0.03 | 2.82±0.03 | 43,058 | 260 |
| GMOOAR-U | **3.02±0.01** | **2.33±0.10** | 2.91±0.02 | 43,685 | 288 |
| GMOOAR-HV | **3.02±0.01** | **2.33±0.09** | **2.92±0.02** | 43,685 | 348 |

Table 1: Average HV, number of parameters and runtime on the multi-task learning datasets.

authors) stores all non-dominated solutions of each iteration in an archive $\mathcal{A}$. On termination, they try all size-$n$ subsets of $\mathcal{A}$ and select the subset with the largest HV on the validation set as the final solution set. As there are $C_n^{|\mathcal{A}|}$ such subsets and $|\mathcal{A}|$ is large when MOO-SVGD terminates, this can be very expensive. In order to be fair to all algorithms being compared, we thus also use the aforementioned commonly practiced strategy on MOO-SVGD.

As in [7], Figure 4 shows the testing performance obtained by the solution set with median HV (over the 10 runs). As can be seen, on Multi-MNIST and Multi-Fashion, the solution sets obtained by COSMOS are dense in the middle but sparse towards the ends, while the solution sets obtained by GMOOAR-U are more uniform. Moreover, compared to all other baselines, the solution sets obtained by GMOOAR are closer to the bottom-right corner where the underlying true PF resides. For MOO-SVGD, many of its obtained solutions are much inferior, and only one of them is in the range shown in Figure 4. A complete plot of all the MOO-SVGD solution sets is in Appendix C.

Table 1 shows the HV's of the solution sets (averaged over the 10 runs), the number of parameters and runtime. As can be seen, GMOOAR has consistently higher HVs than the other baselines. Moreover, its number of parameters is comparable to that of COSMOS, and is much fewer than the other baselines. Compared to the base network LeNet, GMOOAR has only 37% more parameters. In terms of the runtime, GMOOAR is only slightly slower than COSMOS and is much faster than the other baselines. GMOOAR-HV is slower than GMOOAR-U as the computations of HV and its gradient are more expensive [53].

## 4.3 Accuracy-Fairness Trade-off

In this experiment, we follow [44] and aim to achieve both high accuracy and fairness on three tabular datasets: Adult [16], Compass [1], and Default [51]. The accuracy is measured by the cross-entropy loss, while fairness is measured by a hyperbolic tangent relaxation of the Difference of Equality of Opportunity (DEO) [41]. As in [40, 44], a 2-hidden-layer multilayer perceptron is used as the base network. More details can be found in Appendix B.4. The experiment is repeated 10 times.

Figure 5 shows the test loss and fairness measure obtained by the solution set with median HV over the 10 runs. Since the datasets are not difficult, the approximated PFs obtained by various algorithms are close. Solutions obtained by GMOOAR are uniformly distributed, while those obtained by EPO, PHN-LS, PHN-EPO and COSMOS are very dense in the top-left region. Solutions obtained by

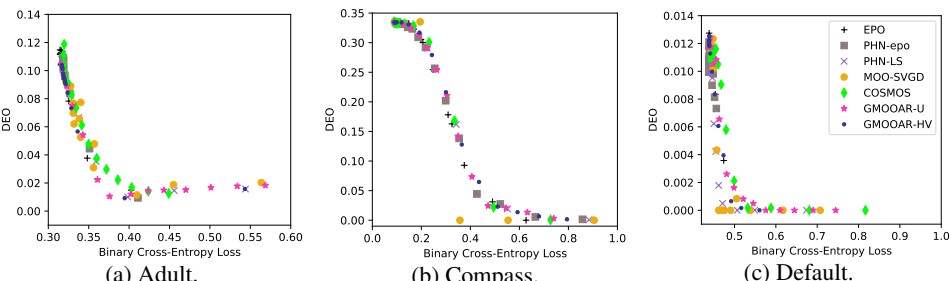

(a) Adult.                    (b) Compass.                    (c) Default.

Figure 5: Test losses and fairness measures obtained on the fairness datasets.

MOO-SVGD are clustered and only cover parts of the PF. Table 2 shows the HV values obtained. It can be seen that GMOOAR achieves better HVs than the baselines.

|  | Adult | Compass | Default |
|---|---|---|---|
| EPO[37] | 3.342±0.001 | 3.709±0.002 | 3.119±0.001 |
| PHN-EPO [40] | 3.340±0.006 | 3.709±0.004 | 3.111±0.005 |
| PHN-LS [40] | 3.341±0.008 | 3.698±0.007 | 3.121±0.003 |
| MOO-SVGD [34] | 3.330±0.008 | 3.716±0.011 | 3.110±0.005 |
| COSMOS [44] | 3.336±0.006 | 3.710±0.004 | 3.114±0.005 |
| GMOOAR-U | 3.344±0.004 | **3.719±0.008** | **3.123±0.004** |
| GMOOAR-HV | **3.345±0.005** | 3.714±0.008 | **3.123±0.002** |

Table 2: Average HV and standard deviation on the fairness datasets.

## 4.4 Larger Networks

To demonstrate that the proposed method can be used on larger networks, we apply GMOOAR on the EfficientNet-B4 [46] with about 17 million parameters. Following [44], we perform experiments on two easy tasks ("Goatee" and "Mustache") and two hard tasks ("Oval Face" and "Pointy Nose") selected from the 40 tasks in CelebA [35]. Since EfficientNet-B4 is around 400 times larger than the LeNet used in previous experiments, PHN-LS, PHN-EPO, and MOO-SVGD cannot be run on our machine. For performance evaluation, the testing cross-entropy loss of each selected task is used. The experiment is repeated 5 times with different random seeds.

Figure 6 shows the test losses obtained by the solution set with median HV over the 5 runs. The corresponding HV values are shown in Table 3. On CelebA-Easy, both GMOOAR-U and GMOOAR-HV outperform COSMOS in terms of HV. On CelebA-Hard, all three algorithms achieve similar HVs, though that of GMOOAR-HV is slightly better.

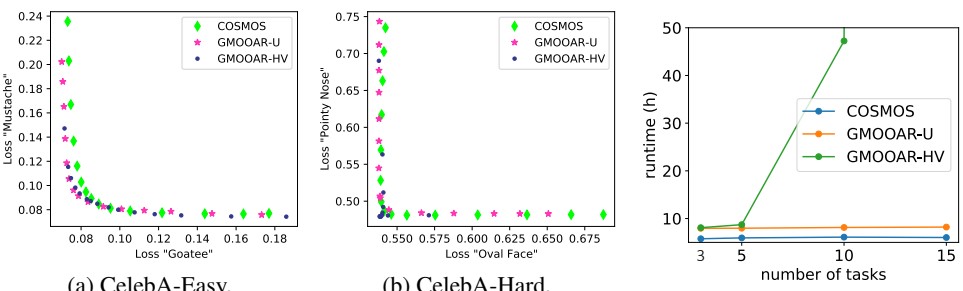

(a) CelebA-Easy.          (b) CelebA-Hard.

Figure 6: Test losses on the two easy tasks (left) and two hard tasks (right) of CelebA.

Figure 7: Average runtime on CelebA with different numbers of tasks.

Next, we select more tasks from CelebA. Table 4 shows the average HV over 3 runs with different random seeds on CelebA with 3, 5, 10, and 15 tasks. Details of the selected tasks are in Appendix B.4. As can be seen, as the number of tasks increases, GMOOAR exhibits more significant performance

|            | CelebA-Easy | CelebA-Hard |
|------------|-------------|-------------|
| COSMOS [44] | 3.700±0.005 | 2.217±0.002 |
| GMOOAR-U | 3.710±0.005 | 2.217±0.002 |
| GMOOAR-HV | **3.711±0.005** | **2.222±0.006** |

Table 3: Average HV and standard deviation of solution sets obtained on CelebA with 2 tasks.

gains compared to COSMOS. In particular, GMOOAR-HV always achieves the highest HV, which is then closely followed by GMOOAR-U. However, the computation of HV is NP-hard with respect to the number of objectives [53]. Hence, as can be seen from Figure 7, the runtime of GMOOAR-HV grows much faster with the number of tasks than COSMOS and GMOOAR-U. On the other hand, GMOOAR-U is very scalable and its running time changes little with the number of tasks.

|            | 3 tasks | 5 tasks | 10 tasks | 15 tasks |
|------------|---------|---------|----------|----------|
| COSMOS [44] | $6.925 \pm 0.005$ | 11.56±0.09 | $118.1 \pm 1.0$ | 1805±52 |
| GMOOAR-U | $6.953 \pm 0.004$ | 15.19±0.18 | $215.0 \pm 6.1$ | 4057±47 |
| GMOOAR-HV | $\mathbf{6.967 \pm 0.007}$ | **15.34±0.06** | $\mathbf{217.8 \pm 1.5}$ | - |

Table 4: Average HV and standard deviation of solution sets obtained on CelebA with 3, 5, 10, and 15 tasks. GMOOAR-HV cannot be run on 15 tasks as it takes more than a month on our machine.

## 4.5 Ablation Study

In this experiment, we study the effects of the RV learning rate $\alpha$ and bandwidth $h$ in (6) on the performance of GMOOAR-U. We use the same setting as in Section 4.3. The experiment is repeated 10 times with different random seeds.

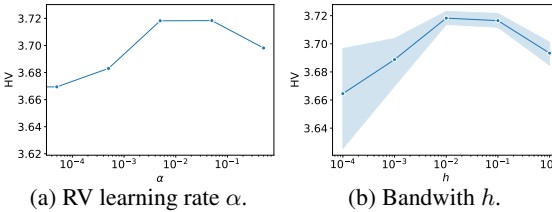

(a) RV learning rate $\alpha$.  (b) Bandwith $h$.

Figure 8: Averge HV and 95% confidence interval with different $\alpha$'s and $h$'s on GMOOAR-U using the Compass dataset.

Figure 8a shows the variations of HV with $\alpha$ ($h$ is fixed to 0.01). As can be seen, an $\alpha$ too small results in almost no RV adaption and thus poor performance, while an $\alpha$ too large may lead to unstable learning. Figure 8b shows the variation of HV with $h$ ($\alpha$ is fixed to 0.005). When $h$ is too small (resp. too large), $Q(R, \phi)$ is close to zero (resp. $n$) and the gradient vanishes, making learning difficult.

## 5 Conclusion

In this paper, we present a novel gradient-based MOO algorithm with adaptive RVs. The proposed algorithm can efficiently adapt the RVs during optimization and provide diverse solutions with a small overhead compared to single-objective optimization. Experiments show the ability of the proposed strategies to obtain well-distributed solutions based on the specified quality function. In the future, we will consider the incorporation of [36] (to obtain a continuous PF) and other state-of-the-art MOO algorithms.

## Acknowledgements

This research was supported in part by the Research Grants Council of the Hong Kong Special Administrative Region (Grant 16200021).

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
