# A Proofs

First, to make the paper self-contained, we state the Lemmas in [23] (with different notations) here.

**Lemma 1.** *[23, Lemma 2] Under Assumption 1, for any $R, \phi$, it holds that*

$$\|\overline{\nabla}_R Q(R, \phi) - \nabla u(\phi)\| \leq L \|\phi^*(R) - \phi\|,$$

*where*

$$L = L_{q,1} + \frac{L_{q,2} C_s}{\mu_s} + C_q \left( \frac{L_{s,1}}{\mu_s} + \frac{L_{s,2} C_s}{\mu_s^2} \right).$$

**Lemma 2.** *[23, Lemma 9] Let $a > 0$, $\{\gamma_j\}_{j \geq 0}$ be a non-increasing, non-negative sequence such that $\gamma_0 < 1/a$. For any $k \geq 0$, it holds that*

$$\sum_{j=0}^{k} \gamma_j \prod_{\ell=j+1}^{k} (1 - \gamma_\ell a) \leq \frac{1}{a}.$$

**Lemma 3.** *[23, Lemma 10] Fix a real number $1 < q \leq 2$. Let $a > 0$, $\{\gamma_j\}_{j \geq 0}$ be a non-increasing, non-negative sequence such that $\gamma_0 < 1/(2a)$. Suppose that $\frac{\gamma_{\ell-1}}{\gamma_\ell} \leq 1 + \frac{a}{2(q-1)} \gamma_\ell$. For any $k \geq 0$, it holds that*

$$\sum_{j=0}^{k} \gamma_j^q \prod_{\ell=j+1}^{k} (1 - \gamma_\ell a) \leq \frac{2}{a} \gamma_k^{q-1}.$$

## A.1 Proof of Theorem 1

We define the following quantities:

$$G_{m:n}^{(1)} = \prod_{i=m}^{n} (1 - \beta_i \mu_s / 4), \quad G_{m:n}^{(2)} = \prod_{i=m}^{n} (1 - \alpha_i \mu_q). \tag{14}$$

First, we introduce the following lemma about the optimality bound for the lower-level objective.

**Lemma 4.** *Under Assumptions 1–2, suppose that the step sizes satisfy the condition in Theorem 1. For any $k \geq 1$, it holds that*

$$\Delta_\phi^{k+1} \leq \prod_{\ell=0}^{k} (1 - \beta_\ell \mu_s / 2) \Delta_\phi^0 + \frac{8}{\mu_s} \left\{ \sigma_s^2 + \frac{4 c_0^2 L_q^2}{\mu_s} \left[ \tilde{\sigma}_q^2 + 3 b_q^2 \right] \right\} \beta_k. \tag{15}$$

*Proof.* The proof is almost the same as in [23, Lemma 3], except that the bias $B_k$ in Assumption 2 is bounded by a constant $b_q$ (instead of a decreasing sequence as in [23]). $\square$

**Lemma 5.** *Under Assumptions 1–2, suppose that the step sizes satisfy the condition in Theorem 1. For any $k \geq 1$, it holds that*

$$\begin{aligned}
\Delta_R^{k+1} &\leq \prod_{\ell=0}^{k} (1 - \alpha_\ell \mu_q) \Delta_R^0 + \left[ \frac{2 \tilde{\sigma}_q^2 + 6 b_q^2}{\mu_q} \right] \alpha_k + \frac{2 b_q^2}{\mu_q^2} \\
&\quad + \left[ \frac{2 L^2}{\mu_d} + 3 \alpha_0 L^2 \right] \sum_{j=0}^{k} \alpha_j \prod_{\ell=j+1}^{k} (1 - \alpha_\ell \mu_d) \Delta_\phi^{j+1}.
\end{aligned} \tag{16}$$

*Proof.* The first few steps below follow [23, Lemma4]. Since $B_k$ in Assumption 2 is bounded by a constant $b_q$ (but not by a decreasing sequence as in [23]), our analysis leads to a different bound.

First, we get the following inequality using the projection property:

$$\|R^{k+1} - R^*\|^2 \leq \|R^k - \alpha_k h_q^k - R^*\|^2 = \|R^k - R^*\|^2 - 2\alpha_k \langle h_q^k, R^k - R^* \rangle + \alpha_k^2 \|h_q^k\|^2.$$

Taking the conditional expectation given $\pi_k$ and using Assumption 2 gives

$$\begin{aligned}
\mathbb{E}[\|R^{k+1} - R^*\|^2 | \pi_k] &\leq \|R^k - R^*\|^2 - 2\alpha_k \langle \nabla u(R^k), R^k - R^* \rangle + \alpha_k^2 \mathbb{E}[\|h_q^k\|^2 | \pi_k] \\
&\quad - 2\alpha_k \langle \overline{\nabla}_R Q(R^k, \phi^{k+1}) - \nabla u(R^k) + B_k, R^k - R^* \rangle.
\end{aligned} \tag{17}$$

Due to strong convexity of $u(R)$, we get

$$\mathbb{E}[\|R^{k+1} - R^*\|^2 | \pi_k]$$

$$\leq (1 - 2\alpha_k \mu_q)\|R^k - R^*\|^2 - 2\alpha_k \langle \bar{\nabla}_R Q(R^k, \phi^{k+1}) - \nabla u(R^k) + B_k, R^k - R^* \rangle + \alpha_k^2 \mathbb{E}[\|h_q^k\|^2 | \pi_k]$$

$$\leq (1 - \alpha_k \mu_q)\|R^k - R^*\|^2 + \frac{\alpha_k}{\mu_q}\|\bar{\nabla}_R Q(R^k, \phi^{k+1}) - \nabla u(R^k) + B_k\|^2 + \alpha_k^2 \mathbb{E}[\|h_q^k\|^2 | \pi_k]$$

$$\leq (1 - \alpha_k \mu_q)\|R^k - R^*\|^2 + (2\alpha_k/\mu_q)\{L^2\|\phi^{k+1} - \phi^*(R^k)\|^2 + b_q^2\} + \alpha_k^2 \mathbb{E}[\|h_q^k\|^2 | \pi_k], \tag{18}$$

where the last inequality is from Lemma 1.

Combining Assumption 2 and Lemma 1 leads to the following inequality:

$$\mathbb{E}_\pi[\|h_q^k\|^2] \leq \tilde{\sigma}_q^2 + 3b_q^2 + 3L^2\|\phi^{k+1} - \phi^*(R^k)\|^2, \tilde{\sigma}_q^2 \equiv \sigma_q^2 + 3\sup_{R \in \mathcal{R}}\|\nabla u(R)\|^2.$$

Substituting the above inequality into (18) and taking the total expectation

$$\Delta_R^{k+1} \leq [1 - \alpha_k \mu_q]\Delta_R^k + [2\alpha_k/\mu_q]L^2\Delta_\phi^{k+1} + 2\alpha_k b_q^2/\mu_q + \alpha_k^2[\tilde{\sigma}_q^2 + 3b_q^2 + 3L^2\Delta_\phi^{k+1}]$$

$$= [1 - \alpha_k \mu_d]\Delta_R^k + [2\alpha_k/\mu_q + 3\alpha_k^2]L^2\Delta_\phi^{k+1} + \alpha_k^2[\tilde{\sigma}_q^2 + 3b_q^2] + 2\alpha_k b_q^2/\mu_q.$$

Then, we obtain the following by recursion

$$\Delta_R^{k+1} \leq G_{0:k}^{(2)}\Delta_R^0 + \sum_{j=0}^k \left\{[\tilde{\sigma}_q^2 + 3b_q^2]\alpha_j^2 G_{j+1:k}^{(2)} + \frac{2b_q^2}{\mu_q}\alpha_j G_{j+1:k}^{(2)} + \left[\frac{2L^2}{\mu_q} + 3\alpha_0 L^2\right]\alpha_j G_{j+1:k}^{(2)}\Delta_\phi^{j+1}\right\}.$$

By applying Lemmas 2 and 3, we get

$$\sum_{j=0}^k \alpha_j G_{j+1:k}^{(2)} \leq \frac{1}{\mu_q}, \quad \sum_{j=0}^k \alpha_j^2 G_{j+1:k}^{(2)} \leq \frac{2\alpha_k}{\mu_q}.$$

Therefore,

$$\Delta_R^{k+1} \leq G_{0:k}^{(2)}\Delta_R^0 + \frac{2}{\mu_q}[\tilde{\sigma}_q^2 + 3b_q^2] \cdot \alpha_k + \frac{2b_q^2}{\mu_q^2} + \left(\frac{2L^2}{\mu_q} + 3\alpha_0 L^2\right)\sum_{j=0}^k \alpha_j G_{j+1:k}^{(2)}\Delta_\phi^{j+1}.$$

$$\square$$

Finally, by substituting (15) into (16), we can get

$$\Delta_R^{k+1} \leq G_{0:k}^{(2)}\left\{\Delta_R^0 + \left[\frac{2L^2}{\mu_q^2} + \frac{3\alpha_0 L^2}{\mu_q}\right]\Delta_\phi^0\right\} + \frac{2}{\mu_q}[\tilde{\sigma}_q^2 + 3b_q^2]\alpha_k + \frac{2b_q^2}{\mu_q^2}$$

$$+ \frac{2\mathrm{c}_1}{\mu_q}\left[\frac{2L^2}{\mu_q} + 3\alpha_0 L^2\right]\frac{8}{\mu_s}\left\{\sigma_s^2 + \frac{4\mathrm{c}_0^2 L_q^2}{\mu_s}[\tilde{\sigma}_q^2 + 3b_q^2]\right\}\alpha_k^{\frac{2}{3}}. \tag{19}$$

The result shown in Theorem 1 is obtained by only considering the dominating terms.

## A.2    Proof of Corollary 1

Recall from (13) that

$$\Delta_\phi^k \lesssim \left[\prod_{i=0}^{k-1}(1 - \beta_i \mu_s/4)\right]\Delta_\phi^0 + \left[\frac{\sigma_s^2}{\mu_s} + \frac{\mathrm{c}_0^2 L_q^2}{\mu_s^2}\tilde{\sigma}_q^2\right]\beta_{k-1}.$$

With diminishing step size $\beta_k = c_\beta/(k + k_\beta)^{2/3}$, it is obvious that $\lim_{k\to\infty}\beta_k \to 0$. Consider the convergence of the first term:

$$\log(\prod_{i=0}^{k-1}(1 - \beta_i \mu_s/4)) = \sum_{i=0}^{k-1}\log(1 - \beta_i \mu_s/4) \leq -\frac{1}{4}\sum_{i=0}^{k-1}\beta_i \mu_s = \frac{c_\beta \mu_s}{4}\sum_{i=0}^{k-1}(i + k_\beta)^{-2/3},$$

where the inequality follows $\log(1-x) = -\sum_{i=1}^\infty x^i/i \leq -x$, for $0 < x < 1$. Since $(i + k_\beta)^{-2/3} \geq (i + k_\beta)^{-1}$ and that the harmonic series is a divergent series, we get $\sum_{i=0}^\infty(i + k_\beta)^{-2/3} = \infty$. Therefore, as $k \to \infty$, $\log(\prod_{i=0}^{k-1}(1 - \beta_i \mu_s/4)) \to -\infty$ and $\prod_{i=0}^{k-1}(1 - \beta_i \mu_s/4) \to 0$. Combining the convergence of the two terms gives $\lim_{k\to\infty}\Delta_\phi^k \to 0$, which means $\phi_k$ converges to the optimal $\phi^*(R_{k-1})$ of (9).

# B Experimental Details

## B.1 Definitions of the Synthetic Problems

The DTZL2 problem [12] minimizes the following $m$ objectives functions with input $x = [x_1, \ldots, x_d] \in \mathbb{R}^d$ (where $0 \leq x_i \leq 1$ for $i = 1, 2, \ldots, d$):

$$f_1(x) = (1 + g(x)) \prod_{i=1}^{m-1} \cos(x_i \pi/2),$$

$$f_2(x) = (1 + g(x)) \sin(x_{m-1}\pi/2) \prod_{i=1}^{m-2} \cos(x_i \pi/2),$$

$$f_3(x) = (1 + g(x)) \sin(x_{m-2}\pi/2) \prod_{i=1}^{m-3} \cos(x_i \pi/2),$$

$$\vdots$$

$$f_m(x) = (1 + g(x)) \sin(x_1 \pi/2),$$

where $g(x) = \sum_{i=m}^{d}(x_i - 0.5)^2$.

The corresponding Pareto front is $\{f^*(x) = [f_1^*(x), \ldots, f_m^*(x)] \in \mathbb{R}^m \mid \sum_{i=1}^{m}(f_i^*(x))^2 = 1, f_i^*(x) \geq 0 \text{ for } i = 1, 2, \ldots, m\}$.

In DTLZ2, the minimum value of each objective is zero, which is usually not the case for most machine learning objectives (e.g., when the loss is used as objective, typically it cannot be minimized to zero). To make the synthetic problems more similar to typical machine learning problems, we apply a linear transform on the objectives $f_i(x)$'s of DTLZ2, as:

$$f_i'(x) = 0.7 f_i(x) + 0.3, \quad i = 1, 2, \ldots, m.$$

We call this Scaled-DTLZ2. Its Pareto front is $\{f^*(x) \mid \sum_{i=1}^{m}((f_i^*(x) - 0.3)/0.7)^2 = 1, f_i^*(x) \geq 0 \text{ for } i = 1, 2, \ldots, m\}$. It can be shown that the optimal value of each objective is nonzero.

MaF1 [8] minimizes the following $m$ objectives with input $x = [x_1, \ldots, x_d] \in \mathbb{R}^d$ (where $0 \leq x_i \leq 1$ for $i = 1, 2, \ldots, d$):

$$f_1(x) = (1 + g(x))(1 - \prod_{i=1}^{m-1} x_i),$$

$$f_2(x) = (1 + g(x))(1 - (1 - x_{m-1}) \prod_{i=1}^{m-2} x_i),$$

$$f_3(x) = (1 + g(x))(1 - (1 - x_{m-2}) \prod_{i=1}^{m-3} x_i),$$

$$\vdots$$

$$f_m(x) = (1 + g(x))(1 - (1 - x_1)),$$

where $g(x) = \sum_{i=m}^{d}(x_i - 0.5)^2$. Its PF is $\{f^* \mid \sum_{i=1}^{m}(1 - f_i^*) = 1, f_i^* \leq 1 \text{ for } i = 1, 2, \ldots, m\}$.

## B.2 Details on Synthetic Problems

We use Adam optimizer [25] and constant learning rates $\alpha = 0.005, \beta = 0.01$. The termination condition is set to 5000 iterations. The penalty strength $\gamma$ in (4) is 5, and the bandwidth $h$ in (6) for GMOOAR-U is 0.02.

## B.3 Reference-Vector-Conditioned Network

For the convolutional neural networks in Sections 4.2 and 4.4, we follow the condition method using the FiLM layer [42] in [15]. Let $x_i$ be the output of layer $i$, and $c$ be the channel index. The

FiLM layer applies a linear transformation to each channel of $x_i$, as: $x'_{i,c} = a_{i,c}x_{i,c} + b_{i,c}$. In the experiment, we generate the FiLM parameters $a, b$ according to reference vector $r$ using an MLP with one hidden layer, i.e., $[a, b] = MLP(r)$. Figure 9 is an illustration of the FiLM layer.

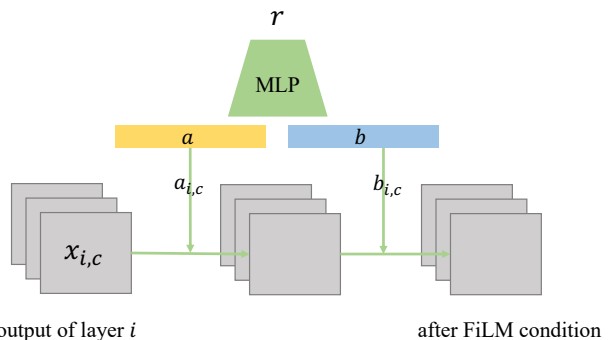

Figure 9: FiLM layer for CNN.

For the LeNet in Section 4.2, we add a FiLM layer after each convolution layer, and the hidden layer dimension of $MLP$ is 20. For the EfficientNet-B4 in Section 4.4, we add a FiLM layer after each of the first four blocks, and the hidden layer dimension of $MLP$ is 20.

On the fairness datasets in Section 4.3, similar to convolutional neural networks, we add a linear transformation layer $x' = ax + b$ after each hidden layer of the MLP. The hidden layer dimension of the $MLP$ for FilM parameter generation is 20.

## B.4 Details on the Real-World Problems

For most hyperparameters, we follow the settings in [44]. Note that we do not perform grid search for hyperparameters. The Adam optimizer[25] is used on all datasets.

**Multi-MNIST, Multi-Fashion, Multi-Fashion+MNIST** We set the initial learning rates $\alpha_1, \beta_1$ to 0.005 and 0.001, respectively. The learning rate is decayed at epochs 20 and 40 by a factor of 0.01. The batch size is 256. We set aside $10\%$ of the training data as the validation set, and perform validation every 5 epochs. Algorithms terminate after 50 epochs. The penalty strength $\gamma$ is 2 for Multi-MNIST and Multi-Fashion, and 8 for Multi-Fashion+MNIST. The bandwidth $h$ for GMOOAR-U is 0.01.

**Adult, Compass, Default** We use an MLP with two hidden layers as the base network. The numbers of hidden units are 60 and 25 for the first and second hidden layers, respectively. We use the constant learning rate $\alpha = 0.005$ and $\beta = 0.001$. The batch size is 256. We use the provided validation set for validation every 5 epochs. Algorithms terminate after 50 epochs. The penalty strength $\gamma$ is set to 0.01 for all three datasets. The bandwidth $h$ for GMOOAR-U is 0.01.

**CelebA** We use the constant learning rate $\alpha = 0.005$ and $\beta = 0.001$. The batch size is 32. We use the provided validation set for validation every 3 epochs. Algorithms terminate after 30 epochs. The penalty strength $\gamma$ is 3, and the bandwidth $h$ for GMOOAR-U is 0.01.

For experiments on CelebA with 3 tasks, we select "Goatee", "Mustache", and "No_Beard" from CelebA as in [44]. For experiments on CelebA with $m \geq 5$ tasks, we select the first $m$ tasks from CelebA:
**CelebA-5**: "5_o_Clock_Shadow", "Arched_Eyebrows", "Attractive", "Bags_Under_Eyes", "Bald"
**CelebA-10**: "5_o_Clock_Shadow", "Arched_Eyebrows", "Attractive", "Bags_Under_Eyes", "Bald", "Bangs", "Big_Lips", "Big_Nose", "Black_Hair", "Blond_Hair"
**CelebA-15**: "5_o_Clock_Shadow", "Arched_Eyebrows", "Attractive", "Bags_Under_Eyes", "Bald", "Bangs", "Big_Lips", "Big_Nose", "Black_Hair", "Blond_Hair", "Blurry", "Brown_Hair", "Bushy_Eyebrows", "Chubby", "Double_Chin"

The total amount of computing for experiments in this paper is around 600 GPU hours on an Nvidia GTX-2080Ti. All the datasets used in this paper are widely-used public datasets without personally identifiable information or offensive content.

## C    Full Results on Multi-Task Learning in Section 4.2

As mentioned in Section 4.2 of the main text, many of the solutions obtained by MOO-SVGD are much inferior, and only one of them is in the range shown in Figure 4 (in the main text). Figure 10 below shows a complete plot on the three datasets in Section 4.2. As can be seen, most of the solutions obtained by MOO-SVGD are dominated by solutions obtained by the other algorithms.

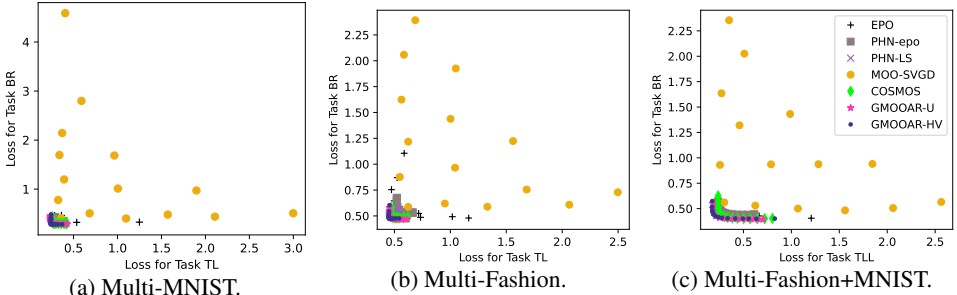

(a) Multi-MNIST.          (b) Multi-Fashion.          (c) Multi-Fashion+MNIST.

Figure 10: BR and TL test losses obtained on the multi-task learning datasets.