# OpenReview forum: "Multi-Objective Deep Learning with Adaptive Reference Vectors"
_NeurIPS.cc/2022/Conference — NeurIPS 2022 Accept_

### Official Review · Reviewer_efXh · 2022-07-08

**Rating:** 5
**Confidence:** 2
**Soundness:** 2 fair
**Presentation:** 3 good
**Contribution:** 2 fair

**Summary:**

This paper propose a novel gradient-based multi-objective optimization (MOO) algorithm with adaptive reference vectors (RVs). They formulate reference vector adaption as a bilevel optimization problem, and solve it with an efficient solver. Experiments on an extensive set of learning scenarios demonstrate the superiority of the proposed algorithm over the state-of-the-art.

**Questions:**

I have some questions about this work:
- As the authors claim in the Line 5 " they may still produce undesirable solutions due to mismatch between the pre-specified reference vectors and the problem’s underlying Pareto front." Then why do we need to get the PF by utilizing the RVs ? I don't understand this point and hope the authors can make some explanations, since this is the major motivation of this work.
- The authors propose to learn the set of RVs simultaneously with the model parameters and formulate the lower-level optimization problem obtains the Pareto-optimal solutions with the given set of RVs, while the upper-level problem optimizes the RVs based on some quality measure. Do the authors try the different level orders? That is the lower-level optimization problem optimizes the RVs while the upper-level optimization problem obtains the Pareto-optimal solutions. I'm just curious about that.

**Limitations:**

For limitations see Weaknesses and Questions. No social impact discussed. Given the effort NeurIPS has put into this effort, I believe something should be added to discuss this.

**Strengths And Weaknesses:**

Strengths:
- The motivation is clear and straightforward.
- This paper is well-written.
- The authors provide theoretical results on the convergence of the proposed Algorithm.
- Experiments on several learning scenarios, including synthetic problems, multi-task learning, accuracy-fairness trade-off, and usage on larger networks demonstrate the effectiveness of the method.

Weaknesses:
- The novelty of this paper is limited. They simply formulate how to generate a proper RV and the model's parameters as a bilevel optimization problem and solve it with an solver, which is limited novelty. Since the idea of RV adaption is commonly used in some evolutionary algorithms [40][26] and the bilevel optimization problem also is a common problem, the authors simply combine these two together to generate their method.
- The solver proposed in this work (GMOOAR) to solve the Bilevel Optimization Problem seems exists some problems. I am not sure can solve the bilevel problem in this simple way, the algorithm needs to be explained more detail.
- How the authors can guarantee the generated RVs are proper RVs?  There needs more experiments to demonstrate this point.
- There miss some SOTA methods to compare, such as *Multi-Objective Meta Learning* (Ye et. al. , NeurIPS 2021).

---

> ### Author Response · Authors · 2022-08-02
> **Responses to Reviewer efXh (1/2)**
>
> Thanks for your recognition and valuable comments. Please see our responses below.
>
> ---
> **Comment 1:** The novelty of this paper is limited. They simply formulate how to generate a proper RV and the model's parameters as a bilevel optimization problem and solve it with an solver, which is limited novelty. Since the idea of RV adaption is commonly used in some evolutionary algorithms [40][26] and the bilevel optimization problem also is a common problem, the authors simply combine these two together to generate their method.
>
> **Response 1:**
> Our main novelties are (1) the use of adaptive reference vectors (RVs), instead of fixed ones as in existing gradient-based MOO algorithms. (2) RV adaptation is based on the gradient of the quality function $Q$, which can generate different distributions according to the decision-maker's preference. Existing algorithms cannot achieve this. (3) The problem is formulated as a bilevel optimization problem that can be efficiently solved.
>
> Although RV adaptation is used in some evolutionary algorithms, their adaptation involves multiple steps to add and remove RVs based on the current population or the archive of all non-dominated solutions. These steps are heuristic, and the balance between exploration and exploitation needs to be carefully considered. Hence, as mentioned in Section 2 and Section 3.2, these evolutionary algorithms are inefficient and cannot be easily extended to gradient-based algorithms. Moreover, these methods cannot provide any theoretical guarantee.
>
> We are the first to use the quality function gradient for RV adaptation. There are two main differences between the proposed method and RV adaptation in evolutionary algorithms: 1) They use the solutions in the current population or external archive to update the RVs; whereas in the proposed gradient-based MOO, we need neither a population nor an archive. We update the RVs using the defined quality function. 2) They update the RVs by adding and removing RVs; while we use the efficient gradient-based method.
>
> In summary, the proposed gradient-based algorithm is entirely different from evolutionary algorithms, and the combination of RV adaptation with bilevel optimization is not trivial.
>
> ---
>
> **Comment 2:** The solver proposed in this work (GMOOAR) to solve the Bilevel Optimization Problem seems exists some problems. I am not sure can solve the bilevel problem in this simple way, the algorithm needs to be explained more detail.
>
> **Response 2:** Bilevel solvers have been used for many well-known problems. The gradient of the lower-level problem can be easily calculated, while the calculation of the gradient of the upper-level problem (hypergradient) is more expensive as it involves backpropagation through the inner loop optimization steps. Therefore, many algorithms (e.g., [1-4]) use a different method to approximate the hypergradient. Here, we adopt the most simple but effective one, i.e., first-order approximation [1][2].
> More details about the gradient calculation and the corresponding approximation error are shown in Section 3.4.
>
> ---
>
> **Comment 3:** How the authors can guarantee the generated RVs are proper RVs? There needs more experiments to demonstrate this point.
>
> **Response 3:**
> RVs are used to weight the different objectives, and so any RV with positive weights is valid. More precisely, if RVs are in the subset $\mathcal{R} \in \mathbb{R}^{m \times n}_+$ (see lines 64-67), they are considered valid. We use the Euclidean projection operator to ensure that RVs are inside $\mathcal{R}$. On the other hand, a predefined
> set of RVs, even though valid, might generate an undesired solution distribution due to the mismatch between the RVs and the Pareto front. But as shown in the experiments, our algorithms can generate a desired solution distribution, which shows that our obtained RVs have a good match with the target Pareto front.

---

> ### Author Response · Authors · 2022-08-02
> **Responses to Reviewer efXh (2/2)**
>
> (Continued)
>
> ---
>
> **Comment 4:** There miss some SOTA methods to compare, such as *Multi-Objective Meta Learning* (Ye et. al. , NeurIPS 2021).
>
> **Response 4:** Our paper aims to find *multiple solutions* for multi-objective single-level optimization problems, while the MOML paper you mentioned can only find *one single solution* for multi-objective *bilevel* optimization problems.
> Hence, they are for different problems and cannot be directly compared. Note that our algorithm can be extended to the MOML setting (i.e., finding multiple solutions for multi-objective bilevel optimization problems).
>
> ---
>
> **Comment 5:** As the authors claim in the Line 5" they may still produce undesirable solutions due to mismatch between the pre-specified reference vectors and the problem's underlying Pareto front." Then why do we need to get the PF by utilizing the RVs ? I don't understand this point and hope the authors can make some explanations, since this is the major motivation of this work.
>
> **Response 5:**
> As a set of *pre-specified* reference vectors may not fit the underlying Pareto front, this motivates us to *adapt* the reference vectors in this paper. There are MOO methods (e.g., MOO-SVGD) that do not utilize RVs entirely. However, RV-based algorithms usually achieve SOTA performance. Also, conditioning using RV can significantly reduce the number of parameters and computation overhead. In Table 1, we show that the proposed algorithm outperforms MOO-SVGD in performance, runtime, and the number of parameters.
>
> ---
>
> **Comment 6:** Do the authors try the different level orders? That is the lower-level optimization problem optimizes the RVs while the upper-level optimization problem obtains the Pareto-optimal solutions. I'm just curious about that.
>
> **Response 6:** Since the optimization of network parameters is based on the RVs, the formulation in our paper is more natural and more consistent with other bilevel optimization problems. For example, in neural architecture search (NAS), the network parameters are optimized in the lower level given the network architecture (learned in the upper level). Similarly, here, we learn the model parameters (in the lower level) given the reference vector (learned in the uppper level).
>
> As suggested by the reviewer, we swap the order of two levels in a new experiment. The average HV obtained by GMOOAR-U (swapped) on Multi-MNITS is 3.0191,  and is almost the same as the original GMOOAR-U (3.0198).
>
> ---
>
> **Comment 7:** No social impact discussed. Given the effort NeurIPS has put into this effort, I believe something should be added to discuss this.
>
> **Response 7:** The proposed algorithms can efficiently provide trade-off solutions between accuracy and fairness, which can promote the machine learning models to be more unbiased to gender, race, or minority groups.
>
> ---
>
> We hope that our responses have addressed the reviewer's concerns. We would be happy to have further discussions.
>
> ---
>
> **References**
>
> [1] Chelsea Finn, Pieter Abbeel, and Sergey Levine. Model-agnostic meta-learning for fast adaptation of deep networks. In International conference on machine learning, pages 1126–1135. PMLR, 2017.
>
> [2] Hanxiao Liu, Karen Simonyan, and Yiming Yang. DARTS: Differentiable architecture search. In International Conference on Learning Representations, 2018.
>
> [3] Rajeswaran, Aravind, Chelsea Finn, Sham M. Kakade, and Sergey Levine. Meta-learning with implicit gradients. Advances in neural information processing systems 32, 2019.
>
> [4] Shaban, Amirreza, Ching-An Cheng, Nathan Hatch, and Byron Boots. Truncated back-propagation for bilevel optimization. In The 22nd International Conference on Artificial Intelligence and Statistics, pp. 1723-1732. PMLR, 2019.

---

### Official Review · Reviewer_bc4u · 2022-07-11

**Rating:** 4
**Confidence:** 4
**Soundness:** 2 fair
**Presentation:** 3 good
**Contribution:** 2 fair

**Summary:**

This paper aims to obtain a set of Pareto-optimal solutions that can better uniformly cover the Pareto front by learning a set of appropriate reference vectors (RVs). The proposed method is formulated as a bilevel optimization problem where we can obtain the Pareto-optimal solution in the inner loop and optimize the RVs by maximizing the quality function in the outer loop. The convergence analysis of the proposed method is provided and empirical studies show the effectiveness of the proposed method.

**Questions:**

Refer to above.

**Limitations:**

Refer to above.

**Strengths And Weaknesses:**

Strengths：

1. This paper aims to obtain a set of Pareto-optimal solutions that can better uniformly cover the Pareto front by learning a set of appropriate reference vectors (RVs), which is meaningful for multi-objective optimization.

2. This paper is well-written and easy to follow.

Weaknesses：

1. The novelty of this paper is limited and it is just an ensemble of the existing works. The RV-conditioned network is adapted from [14] (as mentioned in Section 3.1). The solver (Section 3.3) of the bilevel problem in Eqs. (8)-(9) is more trivial, which is similar to MAML.

2. The description of Pareto domination in Lines 38-39 is problematic.

3. How to compute the gradient w.r.t. $R$ in Eq. (7)?

4. The Eq. (9) is equivalent to minimize $\sum_{j=1}^m (\sum_{i=1}^n r_i(j))f_j(\phi)$ and can only obtain a Pareto-optimal solution based on a new reference vector $(\sum_{i=1}^n r_i(1), \cdots, \sum_{i=1}^n r_i(m))$ rather than a set of Pareto-optimal solutions (as mentioned in Line 120). Thus, how the proposed method can obtain multiple Pareto-optimal solutions and how to guarantee those solutions can uniformly cover the PF.

5. In Section 3.3 and Algorithm 1, it seems the hypergradient (i.e., the gradient of $\phi_{k+1}$ w.r.t. $R$) is ignored, which may cause a large approximation error.

---

> ### Author Response · Authors · 2022-08-02
> **Responses to Reviewer bc4u (1/2)**
>
> Thanks for your helpful comments. Below are our responses to your questions. We hope these can address your concerns.
>
> ---
> **Comment 1:** The novelty of this paper is limited and it is just an ensemble of the existing works. The RV-conditioned network is adapted from [14] (as mentioned in Section 3.1). The solver (Section 3.3) of the bilevel problem in Eqs. (8)-(9) is more trivial, which is similar to MAML.
>
> **Response 1:** Our main novelties are (1) the use of adaptive reference vectors (RVs), instead of fixed ones as in existing gradient-based MOO algorithms. (2) RV adaptation is based on the gradient of the quality function $Q$, which can generate different distributions according to the decision-maker's preference. Existing algorithms cannot achieve this. (3) The problem is formulated as a bilevel optimization problem that can be efficiently solved.
>
> Moreover, note that the conditioned network in [14] is originally designed to reduce the number of parameters, not for RV adaptation. The bi-level optimization solver is not our main novelty. Its choice allows a small computation overhead compared with fixed RVs algorithms but still has theoretical guarantees (Section 3.4).
>
> ---
> **Comment 2:** The description of Pareto domination in Lines 38-39 is problematic.
>
> **Response 2:** Sorry for the confusion. The $\geq$ in line 39 is changed to $>$ in the revised version.
>
> ---
> **Comment 3:** How to compute the gradient w.r.t. $R$ in Eq. (7)?
>
> **Response 3:** We use the algorithm proposed in [1] to compute the gradient of HV. It is based on efficient dimension sweeping algorithms for computing HV.

---

> ### Author Response · Authors · 2022-08-02
> **Responses to Reviewer bc4u (2/2)**
>
> (Continued)
>
> ---
> **Comment 4:** The Eq. (9) is equivalent to minimize $\sum_{j=1}^m (\sum_{i=1}^n r_i(j))f_j(\phi)$ and can only obtain a Pareto-optimal solution based on a new reference vector $(\sum_{i=1}^n r_i(1), \cdots, \sum_{i=1}^n r_i(m))$ rather than a set of Pareto-optimal solutions (as mentioned in Line 120). Thus, how the proposed method can obtain multiple Pareto-optimal solutions and how to guarantee those solutions can uniformly cover the PF.
>
> **Response 4:**
> There might be some confusion due to the use of our symbol $\phi$. In (3), for a particular solution $\phi$, we have
> $$
> s(\phi; r) = \sum_{j=1}^m r(j)f_j(\phi). \tag{a}
> $$
> In (9), we consider a set of $n$ solutions. A straightforward extension of the notation is to use $\{\phi_1,\dots,\phi_n\}$ (line 91). However, as mentioned in Section 3.1, when the solution is obtained by a deep network, $\phi$ corresponds to the network parameters, and using $\{\phi_1,\dots,\phi_n\}$ will lead to a large memory space. Hence, we propose to use a conditioned network $f(\phi,r)$ with network parameter $\phi$. The $i$th solution is generated with input reference vector $r_i$ (line 107) and the corresponding multi-objective value is (line 109)
> $$
> f(\phi,r_i). \tag{b}
> $$
> Now, in (9), as the reviewer noted, we minimize
> $$
> S(\{r_1,\dots,r_n\}, \phi) = \sum_{i=1}^n s(\phi; r_i). \tag{c}
> $$
> With (b), we have to change $s(\phi; r)$ in (a) to $s(\phi; r)=
> \sum_{j=1}^m r(j)f_j(\phi, r)$.
> Substituting into (c), we have
> $$
> S(\{r_1,\dots,r_n\}, \phi)
> = \sum_{i=1}^n \sum_{j=1}^m r_i(j)f_j(\phi, r_i)
> = \sum_{j=1}^m \sum_{i=1}^n r_i(j)f_j(\phi, r_i)
> \neq  \sum_{j=1}^m \left(\sum_{i=1}^n r_i(j)\right) f_j(\phi, r_i)
> $$
> as the reviewer suggested, because the last term also has subscript $i$.
>
> ---
> **Comment 5:** In Section 3.3 and Algorithm 1, it seems the hypergradient (i.e., the gradient of $\phi_{k+1}$ w.r.t. $R$) is ignored, which may cause a large approximation error.
>
> **Response 5:** We use the first-order approximation since the hypergradient computation is expensive, which leads to a large overhead when compared to the based algorithm. This is a common trick, as is used in meta-learning [3] and differentiable neural architecture search [4][5][6],
>
> In the convergence analysis in Section 3.4, we show that the approximation error of such approximation is bounded (Eq. (11)) and the algorithm can converge to the Pareto optimal solutions with such approximation (Corollary 1).
>
> ---
> We hope that our responses have addressed the reviewer's concerns. We would be happy to have further discussions.
>
> ---
>
> **References**
>
> [1] Emmerich, Michael, and André Deutz. "Time complexity and zeros of the hypervolume indicator gradient field." EVOLVE-a bridge between probability, set oriented numerics, and evolutionary computation III. Springer, Heidelberg, 2014. 169-193.
>
> [2] Alexey Dosovitskiy and Josip Djolonga. You only train once: Loss-conditional training of deep networks. In International conference on learning representations, 2019.
>
> [3] Chelsea Finn, Pieter Abbeel, and Sergey Levine. Model-agnostic meta-learning for fast adaptation of deep networks. In International conference on machine learning, pages 1126–1135. PMLR, 2017.
>
> [4] Hanxiao Liu, Karen Simonyan, and Yiming Yang. DARTS: Differentiable architecture search. In International Conference on Learning Representations, 2018.
>
> [5] Dong, Xuanyi, and Yi Yang. Searching for a robust neural architecture in four gpu hours. In Proceedings of the IEEE/CVF Conference on Computer Vision and Pattern Recognition, pp. 1761-1770. 2019.
>
> [6] Ye, Peng, Baopu Li, Yikang Li, Tao Chen, Jiayuan Fan, and Wanli Ouyang. b-DARTS: Beta-Decay Regularization for Differentiable Architecture Search. In Proceedings of the IEEE/CVF Conference on Computer Vision and Pattern Recognition, pp. 10874-10883. 2022.

---

### Official Review · Reviewer_WZqb · 2022-07-12

**Rating:** 7
**Confidence:** 3
**Soundness:** 4 excellent
**Presentation:** 4 excellent
**Contribution:** 3 good

**Summary:**

The paper proposes a novel multi-objective optimization algorithm based on reference vectors (RV).
In contrast to prior works, the paper proposes learning these RVs using gradient-based optimization instead of using pre-specified and fixed RVs.
This is formulated as a bilevel optimization problem, where the lower level obtains the optimal solution given the RVs, and the upper level optimizes a quality measure w.r.t. the learned RVs (uniformity and the hypervolume indicator are considered as quality measures).
An efficient solver for the bilevel optimization problem is proposed, requiring just one gradient descent step for lower and upper-level problems. A convergence analysis is also provided.
The algorithm is evaluated on both synthetic and real-world problems and achieves state-of-the-art performance.

**Questions:**

I would appreciate it if the authors could address the weaknesses mentioned above, i.e.,
- how does the method compare in terms of efficiency with prior approaches
- a discussion of the limitations of the method

**Limitations:**

A section with limitations is unfortunately missing and would definitely benefit the paper.

**Strengths And Weaknesses:**

Strengths:
- The approach to extend existing RV-based approaches with a mechanism to adaptively learn said RVs is sensible and appears to improve upon the baselines in all cases
- The formulation as a bilevel optimization problem is elegant, and the provided solver is efficient, quite simple, and appears to work well in practice.
- The overall presentation is excellent. The technical presentation of the method and background material is sound and clear, and the method is also well-motivated.
- The experiments cover a good number of benchmarks from more artificial/synthetic ones to more real-world applications.

Weaknesses:
- The experiments are most convincing on the somewhat artificial benchmarks around MNIST and FashionMNIST. The experiments might be more compelling with more real-world applications, e.g., multi-label image classification with more than two objectives.
- Adding a comparison in terms of computational efficiency (e.g., in terms of runtime) to Table 1, for example, could be helpful for the readers.
- A discussion of the limitations of the method appears to be missing.

---

> ### Author Response · Authors · 2022-08-02
> **Responses to Reviewer WZqb**
>
> Thanks for your recognition and valuable comments. Please see our responses below.
>
> ---
> **Comment 1:** The experiments are most convincing on the somewhat artificial benchmarks around MNIST and FashionMNIST. The experiments might be more compelling with more real-world applications, e.g., multi-label image classification with more than two objectives.
>
> **Response 1:**
> We use MNIST and FashionMNIST because they are standard benchmark datasets in the MOO literature (such as ParetoMTL[1], EPO[2], PHN[3], COSMOS[4], MOO-SVGD[5]).
> Following the reviewer's suggestion, we also run an additional experiment on the CelebA dataset using three tasks: Goatee, Mustache, and No Beard. The average HV and standard deviation over 3 runs are shown in the table below. The proposed algorithms can still achieve better HV than COSMOS and the HV of GMOOAR-HV is slightly better than that of GMOOAR-U. These results demonstrate that the proposed algorithm can scale to more than two objectives.
>
> | Algorithm | HV |
> | --------- | ----- |
> | COSMOS    | 6.920 $\pm$ 0.003 |
> | GMOOAR-U  | 6.944 $\pm$ 0.004 |
> | GMOOAR-HV | 6.949 $\pm$ 0.011 |
>
> ---
> **Comment 2:** Adding a comparison in terms of computational efficiency (e.g., in terms of runtime) to Table 1, for example, could be helpful for the readers.
>
> **Response 2:** We added the average runtime in Table 1 of the revised paper. The runtime of our algorithm has a small overhead compared to COSMOS and is smaller than the other algorithms. GMOOAR-HV takes a longer time than GMOOAR-U since the computation of HV and HV gradient is more expensive than the quality function that encourages uniformity.
>
> **Comment 3:**  A discussion of the limitations of the method appears to be missing.
>
> **Response 3:**
> A limitation of the current version is that the solutions are discrete. One possible future direction is to incorporate [6] to obtain a continuous PF.
>
> ---
> Thanks again for your suggestions. We would be happy to have further discussions.
>
> ---
> **References**
>
> [1] Xi Lin, Hui-Ling Zhen, Zhenhua Li, Qing-Fu Zhang, and Sam Kwong. Pareto multi-task learning. Advances in neural information processing systems, 32, 2019.
>
> [2] Debabrata Mahapatra and Vaibhav Rajan. Multi-task learning with user preferences: Gradient descent with controlled ascent in pareto optimization. In International Conference on Machine Learning, pages 6597–6607. PMLR, 2020.
>
> [3] Aviv Navon, Aviv Shamsian, Ethan Fetaya, and Gal Chechik. Learning the pareto front with hypernetworks. In International Conference on Learning Representations, 2020.
>
> [4] Michael Ruchte and Josif Grabocka. Scalable pareto front approximation for deep multi366 objective learning. In 2021 IEEE International Conference on Data Mining (ICDM), pages 1306–1311, 2021.
>
> [5] Xingchao Liu, Xin Tong, and Qiang Liu. Profiling pareto front with multi-objective stein variational gradient descent. Advances in Neural Information Processing Systems, 34, 2021.
>
> [6] Pingchuan Ma, Tao Du, and Wojciech Matusik. Efficient continuous pareto exploration in multi-task learning. In International Conference on Machine Learning, pages 6522–6531. PMLR, 2020.

---

### Official Review · Reviewer_5onM · 2022-07-13

**Rating:** 6
**Confidence:** 2
**Soundness:** 4 excellent
**Presentation:** 4 excellent
**Contribution:** 4 excellent

**Summary:**

Paper tackles multi objective optimization (MOO), and proposes a gradient based MOO using reference vectors(RVs). Previous RV based MOO algorithms use a fixed set of RVs, and they might not be able to cover some parts of the Pareto Front (PF). Authors propose to jointly learn RVs along with the solving the MOO problem by formulating it as a bilevel optimization problem. Authors present an algorithm for solving the bilevel optimization problem, and theoretical convergence analysis of the algorithm is also provided.

**Questions:**

See strengths and weaknesses.

**Limitations:**

Authors should add some discussion about whether their algorithm can scale to large scale datasets (for e.g. large scale MTL tasks like image classification + object detection + semantic segmentation ).

**Strengths And Weaknesses:**

The paper is novel and interesting. It seems to be more reasonable to learn the RVs rather than using a fixed set of RVs.

1. Some more details about how the FilM conditioning is done using RVs should be added for completeness.

2. Datasets in Section 4.1 and even the Multi-MNIST, Multi-Fashion and Multi-Fashion+MNIST are synthetically created. It would have been interesting to see results on more real world tasks such as the multi-objective tasks considered in YOTO [14] (image compression).

---

> ### Author Response · Authors · 2022-08-02
> **Responses to Reviewer 5onM**
>
> Thanks for your recognition and valuable comments. Please see our responses below.
>
> ---
> **Comment 1:** Some more details about how the FilM conditioning is done using RVs should be added for completeness.
>
> **Response 1:** Details of the RV-conditioned neural network using FilM are in Appendix B.3. Following the reviewer's suggestion, we have also added a figure for illustration (Figure 9 in Appendix B.3).
>
> ---
>
> **Comment 2:** Datasets in Section 4.1 and even the Multi-MNIST, Multi-Fashion and Multi-Fashion+MNIST are synthetically created. It would have been interesting to see results on more real world tasks such as the multi-objective tasks considered in YOTO [14] (image compression).
>
> Authors should add some discussion about whether their algorithm can scale to large scale datasets (for e.g. large scale MTL tasks like image classification + object detection + semantic segmentation ).
>
> **Response 2:**
> Note that the datasets in Section 4.3 (Adult, Compass, and Default) and Section 4.4 (CelebA) are indeed real-world datasets. Moreover, the CelebA dataset is large-scale with 200K images, which is the largest dataset used in the gradient-based MOO literature.
>
> As the proposed algorithms have a small time and space overhead compared with COSMOS (discussed in Section 3.3), we believe they can scale to even larger datasets. Unfortunately, we could not finish an even larger experiment within the rebuttal period. Instead, the following shows results on the CelebA dataset with 3 tasks: Goatee, Mustache, No Beard. The average HV and standard deviation over 3 runs are shown in the table below. The proposed algorithms still achieve better HV than COSMOS, and the HV of GMOOAR-HV is slightly better than that of GMOOAR-U. These demonstrate that the proposed algorithms can scale to the large-scale dataset with 3 objectives.
>
> | Algorithm | HV |
> | --------- | ----- |
> | COSMOS    | 6.920 $\pm$ 0.003 |
> | GMOOAR-U  | 6.944 $\pm$ 0.004 |
> | GMOOAR-HV | 6.949 $\pm$ 0.011 |
> ---
>
> Thanks again for your suggestions. We would be happy to have further discussions.

---

### Official Review · Reviewer_nfad · 2022-07-14

**Rating:** 5
**Confidence:** 1
**Soundness:** 3 good
**Presentation:** 3 good
**Contribution:** 3 good

**Summary:**

This paper proposed a gradient-based MOO algorithm with adaptive reference vectors. The core idea is to view it as a bilevel optimization problem. Then the authors proposed an interactive solution for this bi-level optimization.

**Questions:**

- It is interesting to view this task as a bi-level optimization problem, but the solver seems not novel. The bi-level solving is similar with existing well-known Expectation–maximization algorithm or the way used in neural architecture search (NAS). So I have concerns about the technical novelty of this paper.

- There are two assumptions proposed in this paper. In which case we should the uniform one (GMOOAR-U) rather than HV-optimal one (GMOOAR-HV)?

- Another concern is the scalability of the proposed method. It can be seem from Table 1 and Table 7 that the performance gain is getting worse and worse when extending to large models.



**Limitations:**

The authors did not talk about limitations of their work.

**Strengths And Weaknesses:**

Strength:
- A new way to reformulate the task to bi-level optimization
- Better performance than state-of-the-art methods.


Weakness:
- It is interesting to view this task as a bi-level optimization problem, but the solver seems not novel. The bi-level solving is similar with existing well-known Expectation–maximization algorithm or the way used in neural architecture search (NAS). So I have concerns about the technical novelty of this paper.

- There are two assumptions proposed in this paper. In which case we should the uniform one (GMOOAR-U) rather than HV-optimal one (GMOOAR-HV)?

- Another concern is the scalability of the proposed method. It can be seem from Table 1 and Table 7 that the performance gain is getting worse and worse when extending to large models.

---

> ### Author Response · Authors · 2022-08-02
> **Responses to Reviewer nfad**
>
> Thanks for your helpful comments. Below are our responses to your questions. We hope these can address your concerns.
>
> ---
> **Comment 1:** It is interesting to view this task as a bi-level optimization problem, but the solver seems not novel. The bi-level solving is similar with existing well-known Expectation–maximization algorithm or the way used in neural architecture search (NAS). So I have concerns about the technical novelty of this paper.
>
> **Response 1:** The bi-level optimization solver is not our main novelty. Instead, our main novelties are: (1) the use of adaptive reference vectors (RVs), instead of fixed ones as in existing gradient-based MOO algorithms. (2) RV adaptation is based on the gradient of the quality function $Q$, which can generate different distributions according to the decision-maker's preference. Existing algorithms cannot achieve this. (3) The problem is formulated as a bilevel optimization problem that can be efficiently solved.
>
> The bilevel solver is just a tool to solve the proposed novel problem. A lot of well-known algorithms use bilevel solvers for different problems (e.g., MAML[1] for meta-learning, DARTS [2] for NAS). Also, the EM algorithm is usually not considered bilevel optimization.
>
> Besides, we use the first-order approximation for the upper-level gradient in the proposed solver. It has a small computation overhead compared with fixed-RV algorithms but still has theoretical guarantees (Section 3.4).
>
> ---
> **Comment 2:** There are two assumptions proposed in this paper. In which case we should the uniform one (GMOOAR-U) rather than HV-optimal one (GMOOAR-HV)?
>
> **Response 2:**
> GMOOAR-U and GMOOAR-HV do not correspond to different assumptions. Instead, they correspond to different choices of the quality function (which in turn depends on the preference of the decision-maker). The quality function of GMOOAR-U encourages the solutions obtained to be more uniformly distributed, while the quality function of GMOOAR-HV encourages the set of solutions obtained to have higher HV (which is a popular measure in MOO, as mentioned in line 49).
>
> ---
> **Comment 3:** Another concern is the scalability of the proposed method. It can be seem from Table 1 and Table 7 that the performance gain is getting worse and worse when extending to large models.
>
> **Response 3:**
> An important goal of the paper is to obtain diverse and representative trade-off solutions. Hence, one should not only look at the HV values shown in those tables. For example, from Table 2, the difference in HV between COSMOS and GMOOAR-U is 0.009 on the compass dataset, which appears small. However, from Figure 5(b), we can see that there is a significant difference in solution distribution. Specifically, the COSMOS solutions are very dense in the top-left region, while the GMOOAR-U solutions are much more uniform. Similarly, the obtained solutions on the larger model (Figure 6) can still achieve preferred distributions.
>
> ---
> **Comment 4:** The authors did not talk about limitations of their work.
>
> **Response 4:**
> A limitation of the current version is that the solutions are discrete. One possible future direction is to incorporate [3] to obtain a continuous PF.
> ---
>
> We hope that our responses have addressed the reviewer's concerns. We would be happy to have further discussions.
>
> ---
> **References**
>
> [1] Chelsea Finn, Pieter Abbeel, and Sergey Levine. Model-agnostic meta-learning for fast adaptation of deep networks. In International conference on machine learning, pages 1126–1135. PMLR, 2017.
>
> [2] Hanxiao Liu, Karen Simonyan, and Yiming Yang. DARTS: Differentiable architecture search. In International Conference on Learning Representations, 2018.
>
> [3] Pingchuan Ma, Tao Du, and Wojciech Matusik. Efficient continuous pareto exploration in multi-task learning. In International Conference on Machine Learning, pages 6522–6531. PMLR, 2020.

---

> > ### Comment · Reviewer_nfad · 2022-08-09
> > **Thanks for the response.**
> >
> > Thanks for the response and it addressed my prior concerns. I am not an expert in this area, but I increased my rating.

---

### Meta-Review · Area_Chair_8qxV · 2022-08-26

**Recommendation:** Accept
**Confidence:** Less certain

**Metareview:**

The paper tackles the multi-objective optimization (MOO) problem using adaptive reference vectors(RVs). The authors claim two main contributions:  (1) the use of adaptive reference vectors (RVs), instead of fixed ones as in existing gradient-based MOO algorithms； (2) RV adaptation is based on the gradient of the quality function.  It is reasonable to learn the RVs rather than using a fixed set of RVs. The authors also provide theoretical results about the proposed method. However, the concerns raised by reviewers should be carefully considered and addressed. Also, the authors also consider only 2 or 3 objectives. It would be interesting to see results with more objectives  (see >10) on classical numerical multi-objective optimization problems.  Last, Figure 1 is confusing and difficult to understand.

**Award:**

No

---

### Decision · Program_Chairs · 2022-09-14

Accept